🔓 | **Open Peer Review** | Bacteriology | Research Article

# Metabolic and transcriptional activities underlie stationary-phase *Pseudomonas aeruginosa* sensitivity to Levofloxacin

Patricia J. Hare,[1,2] Juliet R. Gonzalez,[1] Ryan M. Quelle,[1] Yi I. Wu,[3] Wendy W. K. Mok[1]

**ABSTRACT** The ubiquitous opportunistic pathogen *Pseudomonas aeruginosa* is highly adaptive and refractory to several different classes of antibiotics. However, we found in this study that stationary-phase *P. aeruginosa* cultures exhibit greater sensitivity to the fluoroquinolone Levofloxacin (Levo) than other bactericidal antibiotics, including an aminoglycoside (Tobramycin) and β-lactam (Aztreonam). To understand the basis of this sensitivity, we conducted time-lapse fluorescence microscopy experiments of cells during Levo treatment. We discovered that stationary-phase *P. aeruginosa* cells die rapidly during treatment and undergo heterogeneous morphological changes, including explosive lysis, filamentation, and gradual loss of membrane integrity as evidenced by propidium iodide uptake. These morphologies are reminiscent of how the model organism *Escherichia coli* appears when recovering from fluoroquinolone treatment, a period when activation of the DNA damage-induced SOS response is crucial. Accordingly, we monitored the morphologies and survival of *P. aeruginosa* Δ*recA* mutants and found that the SOS response is not involved in *P. aeruginosa* Levo sensitivity like it is for *E. coli*. We hypothesized that Levo sensitivity may be due to *P. aeruginosa* maintaining active metabolism in stationary phase. We determined that stationary-phase *P. aeruginosa* cells transcribe, maintain reductase activity, and accumulate reactive metabolic species which contribute to Levo-mediated death. By elucidating how *P. aeruginosa* cells sustain metabolic activity during the stationary phase, we can design strategies to sensitize these persistent subpopulations to Levo and maintain the efficacy of this clinically important fluoroquinolone antibiotic.

**IMPORTANCE** The bacterial pathogen *Pseudomonas aeruginosa* is responsible for a variety of chronic human infections. Even in the absence of identifiable resistance mutations, this pathogen can tolerate lethal antibiotic doses through phenotypic strategies like biofilm formation and metabolic quiescence. In this study, we determined that *P. aeruginosa* maintains greater metabolic activity in the stationary phase compared to the model organism, *Escherichia coli*, which has traditionally been used to study fluoroquinolone antibiotic tolerance. We demonstrate that hallmarks of *E. coli* fluoroquinolone tolerance are not conserved in *P. aeruginosa*, including the timing of cell death and necessity of the SOS DNA damage response for survival. The heightened sensitivity of stationary-phase *P. aeruginosa* to fluoroquinolones is attributed to maintained transcriptional and reductase activity. Our data suggest that perturbations that suppress transcription and respiration in *P. aeruginosa* may actually protect the pathogen against this important class of antibiotics.

**KEYWORDS** *Pseudomonas aeruginosa*, fluoroquinolone, antibiotic tolerance, transcription, stationary phase

Address correspondence to Wendy W. K. Mok, mok@uchc.edu.

The authors declare no conflict of interest.

See the funding table on p. 14.

*P*seudomonas aeruginosa is a ubiquitous pathogen responsible for a variety of human infections that are often chronic and antibiotic-refractory (1, 2). Even in the absence

of identifiable resistance mutations, this pathogen can tolerate lethal antibiotic doses by undergoing reversible phenotypic changes, as observed for bacteria in biofilms (3–5). On a single-cell level, stochastic decreases in membrane permeability or increases in efflux capacity can limit effective antibiotic concentrations to facilitate the survival of persistent subpopulations of cells (6–11). Another key determinant of antibiotic effectiveness in susceptible populations is cellular metabolism (12–14). In *P. aeruginosa*, metabolic heterogeneity due to differential nutrition and oxygenation has been shown to impact antibiotic tolerance (11, 15–18).

The connection between metabolism and antibiotic tolerance is most salient in the contrast between exponential-phase and stationary-phase bacterial cultures. Rapidly growing cells generally have lower survival during treatment with bactericidal antibiotics than quiescent cells in stationary phase (3, 19). However, many studies on phenotypic responses to antibiotics have been conducted with logarithmically growing cells or stationary-phase cultures that are resuspended in fresh, nutrient-rich media during drug treatment (11, 20, 21). These conditions are insufficient to model how pathogens exist in infection sites, where bacterial growth is likely limited by host stresses and nutrient availability (22, 23).

Foundational work on antibiotic tolerance and persistence has primarily been focused on the lab-adapted *E. coli* strain MG1655 and its derivatives (7, 13, 24–27). While these studies have generated fundamental insight into persister physiology, lab-adapted *E. coli* strains have differing metabolic potential/requirements compared to *P. aeruginosa*, which evolved to survive and colonize various environments and host niches. In addition to a battery of virulence factors and quorum sensing molecules that *E. coli* MG1655 does not produce, *P. aeruginosa* secretes redox-active phenazines that contribute to its multipurpose lifestyle in aerobic and anaerobic environments (28). Additionally, *P. aeruginosa* preferentially consumes the tricarboxylic acid cycle intermediate, succinate, over glucose (29). Therefore, the metabolic programs that underlie *E. coli* antibiotic tolerance/persistence may not apply to this related—but clearly distinct—species.

In this study, we show that stationary-phase *P. aeruginosa* strains PAO1 and PA14 retain metabolic activity that greatly surpasses that of *E. coli* MG1655. This metabolic activity sensitizes *P. aeruginosa* to the DNA topoisomerase-inhibiting fluoroquinolone Levofloxacin (Levo), but not to other classes of bactericidal antibiotics. The finding that fluoroquinolones can kill stationary-phase *P. aeruginosa*, often more effectively than aminoglycoside antibiotics, is consistent with previous literature (10, 30–33). We analyzed the metabolic states of *P. aeruginosa* and determined that fluoroquinolone sensitivity is most likely explained by the cells' active transcription during the stationary phase. Furthermore, we discovered unexpected deviations from hallmarks of *E. coli* fluoroquinolone persistence—namely, the dependence on the SOS response, morphological changes during Levo treatment, and timing of cell death. Altogether, this work sheds new light on the metabolic state of stationary-phase *P. aeruginosa* and how it contrasts the responses to fluoroquinolones that have previously been described in *E. coli*.

## RESULTS

### Stationary-phase *P. aeruginosa* is more sensitive to Levo than Tobramycin or Aztreonam

Bactericidal antibiotics retain some activity against stationary-phase cultures, whereas bacteriostatic drugs are generally less effective against nongrowing cells (20, 32–34). We began by asking whether the persistence of stationary-phase cultures of *P. aeruginosa* would differ against bactericidal antibiotics with different primary targets. For generalizability, we tested *P. aeruginosa* strains PAO1 and PA14—representing the two major genetic clades of this species—in each experiment (35). We chose antibiotics representing the three major classes of bactericidal antibiotics—Aztreonam (β-lactams), Tobramycin (aminoglycosides), and Levo (fluoroquinolones)—based on their utilization in the treatment of clinical pseudomonal infections (36). After confirming each strain's

susceptibility to the antibiotics (Fig. 1A; Fig. S1; Table S1), we performed survival assays at supra-MIC concentrations to determine how sensitive stationary-phase PAO1 and PA14 were to each antibiotic, relative to the MICs (Fig. 1B through D). Stationary-phase cultures were grown in chemically defined media with succinate as the sole carbon source before treatment. We found that *P. aeruginosa* was completely tolerant to Tobramycin at 15× MIC and Aztreonam up to 25–50× MIC. Comparatively, *P. aeruginosa* was highly sensitive to Levo; we detected approximately six-log decreases in cell survival when Levo was administered at 15× MIC (5 µg/mL).

## Stationary-phase *P. aeruginosa* dies during Levo treatment

To begin understanding why stationary-phase *P. aeruginosa* is so sensitive to Levo, we monitored cells during the course of antibiotic treatment using time-lapse fluorescence microscopy. Cells were grown to stationary phase in planktonic cultures for 16 h in the presence of SytoxBlue, a semi-permeable dye that can only stain cells with compromised membrane integrity. Cells from these stationary-phase cultures were diluted 30-fold in PBS and seeded onto agarose pads prepared with the culture's own cell-free spent media along with 5 µg/mL Levo (~15× MIC) and propidium iodide (PI) to stain for non-viable cells during antibiotic treatment. Any cells that were SytoxBlue-positive were classified as having died during the growth to stationary phase, before seeding onto the agarose pad, whereas PI-positive cells were classified as having died after exposure to Levo in the agarose pad. We confirmed that SytoxBlue does not affect cultures' ability to grow to stationary phase (Fig. S2A through C) and that the concentration of PI used for imaging does not cause cell death itself (Fig. S2D through F).

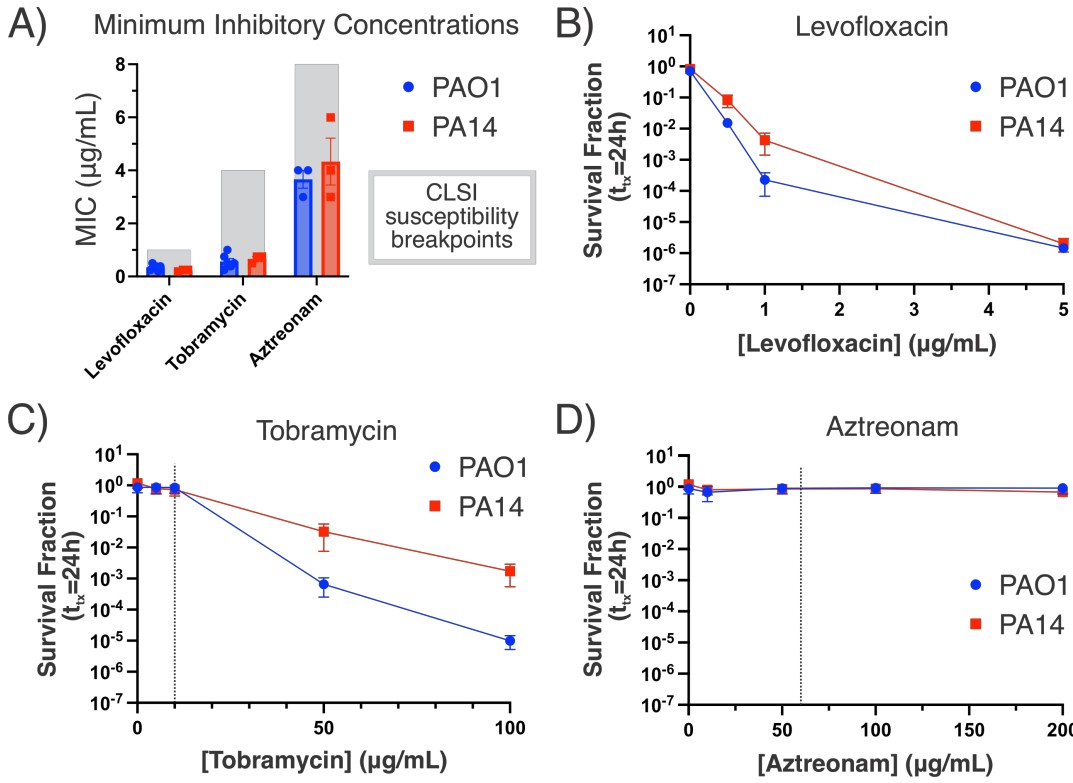

**FIG 1** Stationary-phase *P. aeruginosa* is susceptible to Levo. (A) Minimum inhibitory concentrations of *P. aeruginosa* PAO1 and PA14, as measured by testing growth inhibition of low-density cultures with MIC test strips (*n* ≥ 3). The gray boxes denote the CLSI MIC susceptibility breakpoints for *P. aeruginosa* for the respective antibiotics. Drug concentration-dependent killing of stationary-phase *P. aeruginosa* against Levo (B), Tobramycin (C), or Aztreonam (D). The vertical dashed lines represent ~15× MIC for the respective drugs. Survival fractions are calculated as the colony-forming units per mL (CFU/mL) after 24 h treatment at the given antibiotic concentration divided by the CFU/mL at the time of treatment for each strain. Data represent the mean and standard error of the mean for three biological replicates (*n* = 3).

Over 24 h treatment, *P. aeruginosa* undergoes a variety of morphological changes, including filamentation, septation, and spheroplast formation preceding explosive cell lysis (Fig. 2A; Video S1 to S4). These changes drastically differ from the response of the model Gram-negative bacterium, *E. coli*, to Levo (Fig. S3; Video S5). Consistent with previous studies, we found that stationary-phase *E. coli* MG1655 endured Levo treatment without noticeable changes in cell morphology and only lose viability later in treatment (25, 37). Our quantifications of cell morphology (intact or lysed) and PI-positivity demonstrate how much more responsive stationary-phase *P. aeruginosa* is to Levo compared to *E. coli* (Fig. 2B through D). Fewer than 1% of cells were SytoxBlue-positive, so there was negligible loss of viability during 16 h pregrowth (Fig. S4). However, a large fraction of *P. aeruginosa* (~40%) was PI-positive at the start of imaging. Because cells were

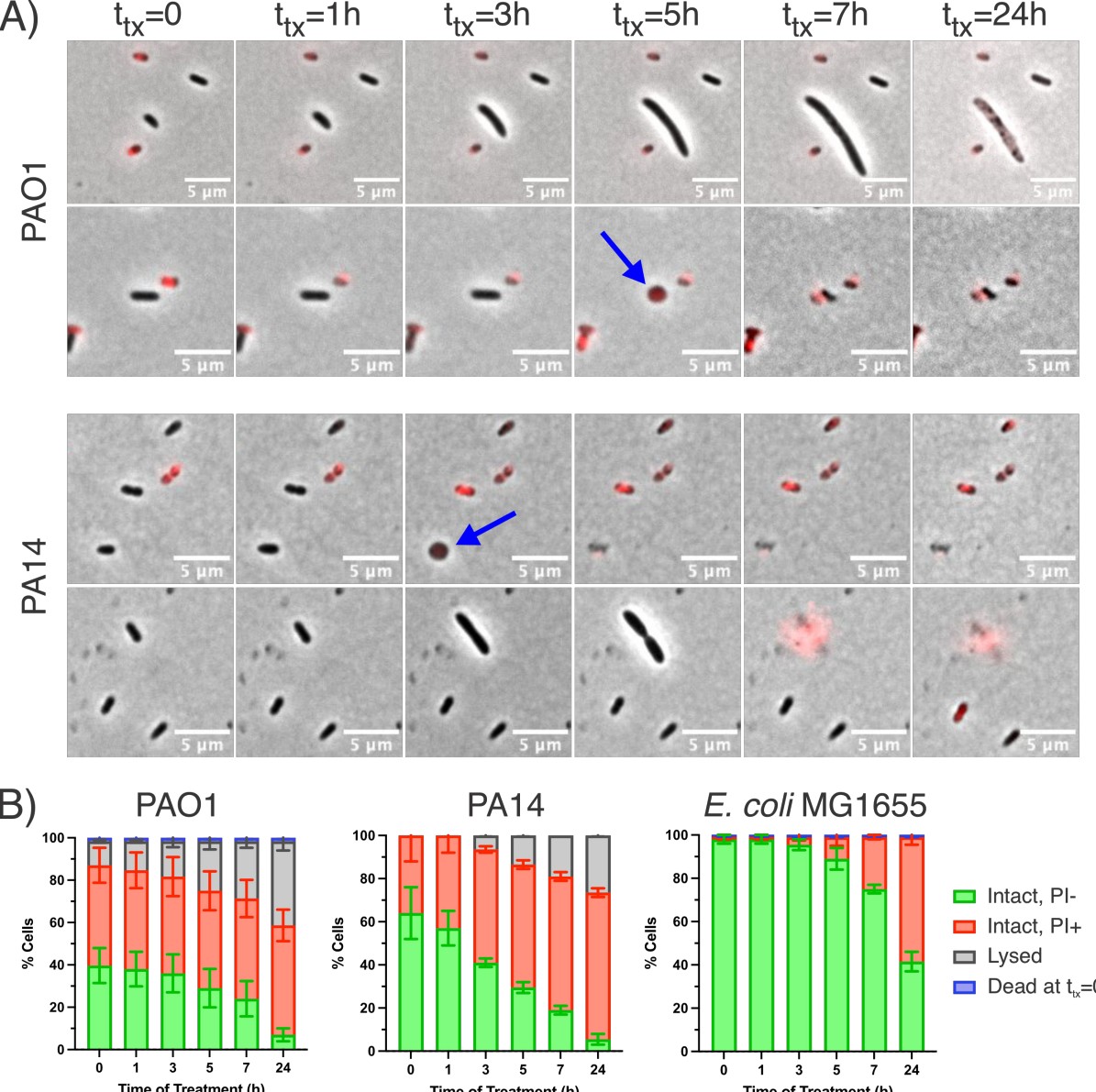

**FIG 2** Stationary-phase *P. aeruginosa* dies during Levo treatment. (A) Representative images of *P. aeruginosa* PAO1 and PA14 during treatment [merged phase and red fluorescent (for PI detection) channels]. The blue arrows indicate cells that have formed spheroplasts. Two sets of images are shown for each strain. (B) Quantification of cell statuses during time-lapse microscopy experiments of PAO1, PA14, and *E. coli* MG1655 throughout Levo treatment. Cells were classified by their morphology (intact or lysed), uptake of PI (PI+ indicates non-viability), or staining for SytoxBlue (designated as dead at $t_{tx} = 0$). Data represent the averages from three biological replicates ($n = 300$ cells) for PAO1 and two biological replicates ($n = 200$ cells) for PA14 and *E. coli* MG1655.

not SytoxBlue-positive, we can conclude that they died rapidly in the ~30 min between being seeded onto the agarose pad and the start of time-lapse imaging.

Lysis during genotoxic stress may be attributed to the induction of prophages (38, 39). To determine whether phages were responsible for the observed lytic phenotypes, we collected cell-free spent media from untreated or Levo-treated cultures to use in plaque assays. We reasoned that if cells in Levo-treated cultures were lysed due to the release of lysogenized phages, then the spent media would contain those lytic phages and lyse untreated *P. aeruginosa* in the plaque assays. The phages JBD90 and JBD23 are known to lyse PAO1 and PA14, respectively, and formed plaques as expected (40); however, the spent media collected from Levo-treated populations were insufficient for plaque formation on either PAO1 or PA14 (Fig. S5). Taken together, these data suggest that *P. aeruginosa* lysis and sensitivity to Levo are not attributable to prophage activation during Levo treatment.

## Loss of RecA/SOS response does not impact *P. aeruginosa* Levo survival

The morphological changes of *P. aeruginosa* during Levo treatment are reminiscent of the lysis and filamentation that stationary-phase *E. coli* undergoes after Levo treatment (25, 37, 41). As fluoroquinolone-treated *E. coli* recovers on antibiotic-free agarose pads containing nutrients, the majority of cells filament due to induction of the SOS response, which leads to inhibition of cell division via SulA (25, 37, 41). We therefore asked whether deleting *recA*, which encodes a key protein in the SOS response regulon, would abrogate the morphological changes we observed in *P. aeruginosa* during treatment. Δ*recA* strains were constructed by allelic exchange in wild-type PAO1 and PA14. The knockouts were verified by whole-genome sequencing (WGS) and were found to have lower Levo MICs than the parental wild-type strains, which is consistent with the literature (Fig. S6) (42, 43). During Levo treatment, PAO1 Δ*recA* and PA14 Δ*recA* still lengthen, but they do not form spheroplasts or lyse explosively (Fig. 3A and B; Video S6 to S9). These data corroborate previously described phenotypes of Ciprofloxacin-treated wild-type and *recA* deletion strains as outcomes of programmed cell death pathways downstream of the SOS response (44–46). Unexpectedly, we did not observe statistically significant differences between the survival of stationary-phase *P. aeruginosa recA* knockouts and their wildtype counterparts across the range of Levo concentrations that we tested (Fig. 3C). This suggests that *recA* is not necessary for Levo persistence in stationary-phase *P. aeruginosa* like it is *E. coli* (47, 48). In sum, *recA* affects the way in which stationary-phase *P. aeruginosa* cells die, but it does not govern how much of the population ultimately succumbs to Levo treatment.

Altogether, the data suggest that a shared mechanism upstream of the SOS response drives the sensitivity of *P. aeruginosa* strains to Levo. We hypothesized that stationary-phase *P. aeruginosa* retains metabolic activity that leads to death during Levo treatment and renders it more sensitive to Levo's action overall.

## Depth of stationary phase does not increase Levo persistence

We performed due diligence to demonstrate that 16 h incubation in chemically defined Basal Salt Medium (BSM) with succinate as the sole carbon source is sufficient for PAO1 and PA14 to reach stationary phase. By 16 h, cultures reach a maximum optical density (Fig. S7A) and deplete the available succinate to below the limit of detection (Fig. S7B). The population's stagnation and succinate consumption indicate that the cultures are in stationary phase.

We then asked whether *P. aeruginosa*'s sensitivity to Levo in our experiments was due to insufficiently quiescent cultures (49). We repeated our persister assays with cultures that were "deeper" in stationary phase to determine if these cultures exhibit increased survival against Levo. When we extended the duration of culture growth before Levo treatment, we found that cultures grown for 24 or 48 h had comparable survival against Levo as the standard 16 h stationary-phase cultures, thus refuting the possibility that cultures were insufficiently growth-arrested as an explanation for their

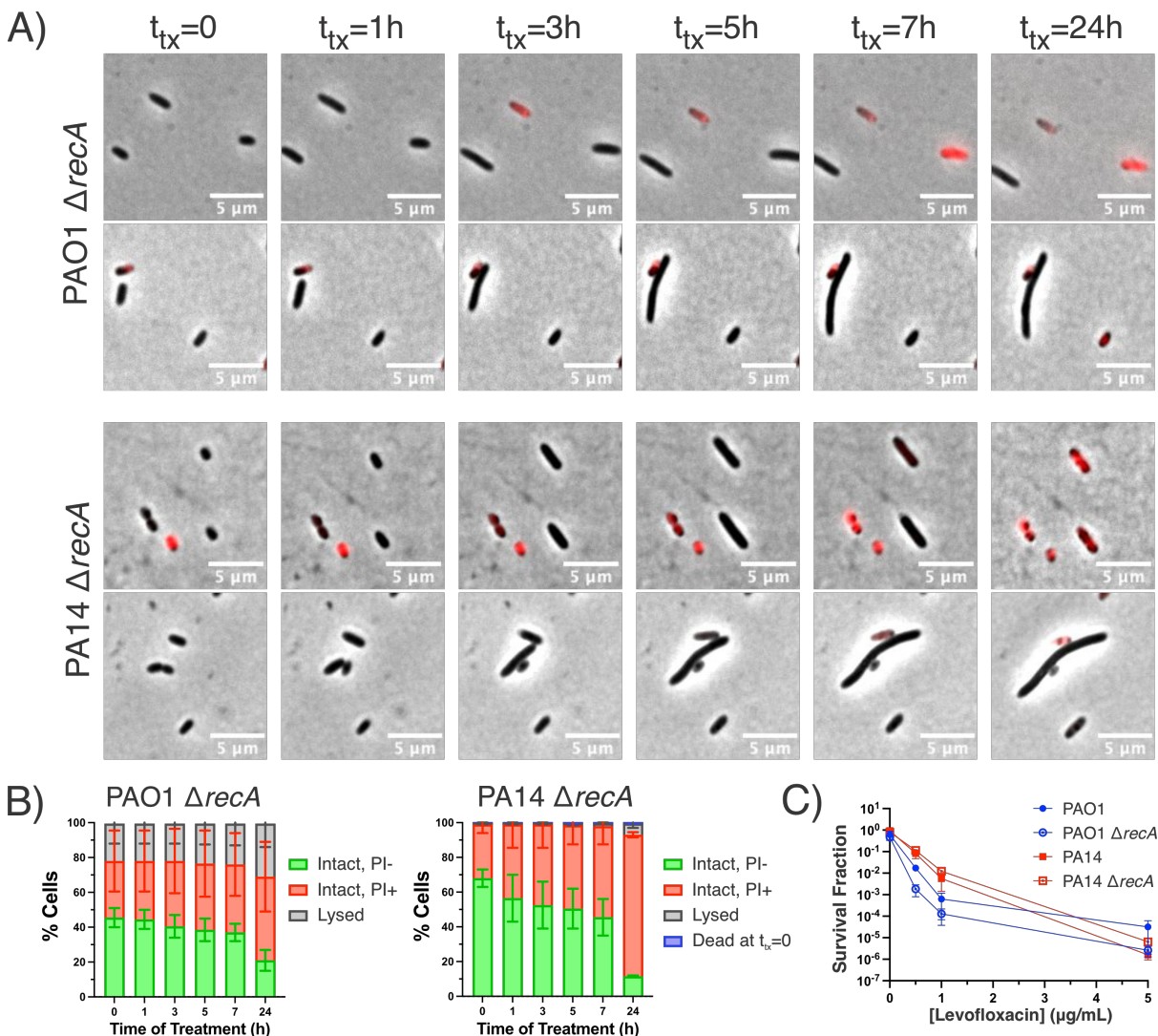

FIG 3  Loss of RecA impacts morphological changes in *P. aeruginosa* during Levo treatment, but not survival. (A) Representative images of *P. aeruginosa* PAO1 Δ*recA* and PA14 Δ*recA* during Levo treatment. (B) Quantification of cell fates during time-lapse microscopy experiments of PAO1 Δ*recA* and PA14 Δ*recA* throughout Levo treatment. Cells were classified by their morphology (intact or lysed) and uptake of PI (PI+ indicates non-viability). PA14 Δ*recA* cells were also classified by their staining for SytoxBlue (designated as dead at $t_{tx}$ = 0). Data represent the averages from 200 cells across two biological replicates for each strain. (C) Δ*recA* strains have similar Levo susceptibility as their wild-type counterparts after 24 h treatment [*n* = 3; not statistically significant (*P* > 0.05) by Mann-Whitney test with multiple test correction].

Levo sensitivity (Fig. 4A and B). From these data, we hypothesized that *P. aeruginosa* maintains a nongrowing yet metabolically active state over several days of culture, which confers sensitivity to Levo in the stationary phase.

## Stationary-phase *P. aeruginosa* maintains reductase activity

We began exploring the metabolic activity of stationary-phase *P. aeruginosa* with Redox Sensor Green (RSG), a dye that becomes fluorescent when reduced by cellular reductases. The primary contributors to reducing activity in cells are the reductases of the electron transport chain during cellular respiration (50, 51). Our data demonstrate that *P. aeruginosa* PAO1 and PA14 have reducing activity even after 48 h in culture and that these levels are comparable to exponential-phase cultures (Fig. 5A). In contrast, *E. coli* MG1655 stained with RSG is only fluorescent in exponential-phase cultures, not 16 h stationary-phase cultures (Fig. S8).

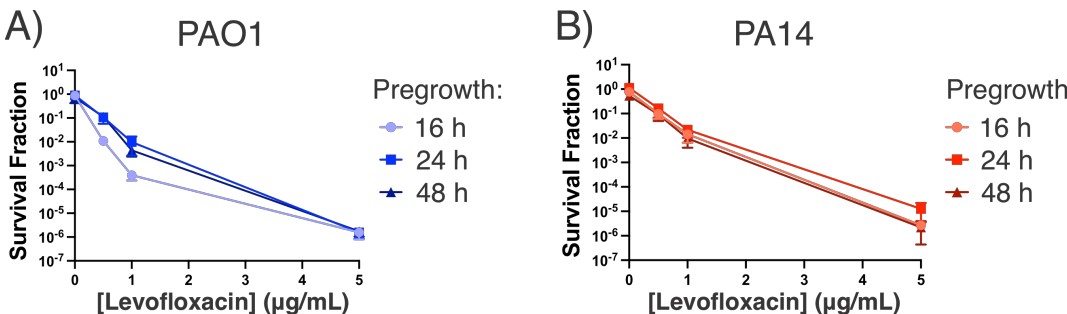

**FIG 4** Culturing *P. aeruginosa* for longer periods before Levo treatment does not alter its survival against Levo. (A) PAO1 and (B) PA14 were grown for 16, 24, or 48 h before treatment with different concentrations of Levo and assessment of survival after 24 h treatment by colony counts. Data represent the mean and standard error of the mean for three biological replicates.

As negative controls, we treated cells with carbonyl cyanide m-chlorophenyl hydrazine (CCCP), an ionophore that dissipates the proton gradient and disrupts cellular respiration, before RSG was added. CCCP-treated cells had negligible RSG signal, supporting the interpretation that RSG fluorescence is a result of electron transport chain activity. A possible confounder that might reduce RSG is the redox-active phenazines that *P. aeruginosa* produces (28). For example, pyocyanin is a phenazine that acts as a terminal electron acceptor in anoxic conditions, but it can also serve as an electron donor depending on the recipient molecule's redox potential. To test whether phenazines or other components of the culture media were reducing RSG, we incubated the cell-free conditioned media from stationary-phase *P. aeruginosa* cultures with RSG. We found that there was no increase in RSG fluorescence (Fig. S9). Based on these data, we conclude that the RSG fluorescence we observe for our samples is due to electron flux within cells and not extraneous media components.

We reasoned that, if cells are continuing respiration into the stationary phase, Levo treatment could disrupt metabolism and increase the production of reactive metabolites, which can contribute to cell death (52, 53). We asked whether adding an antioxidant (thiourea) or iron chelator (2,2'-bipyridine) during Levo treatment could rescue *P. aeruginosa* by decreasing the amount of oxidative stress. Thiourea significantly increased the survival of cells treated with Levo by 24 h for both PAO1 and PA14 (Fig. 5B); however, iron chelation only improved Levo survival for PAO1 but not for PA14 (Fig. 5C). We note that we utilized thiourea at its MIC (Table S4) in accordance with concentrations used in previous studies (54). Therefore, the fact that thiourea could rescue Levo-treated cells instead of lowering survival suggests that its antioxidant properties outweighed potentially inhibitory effects.

Our RSG fluorescence data suggest that *P. aeruginosa* maintains substantial reductase activity in the stationary phase and that continued electron flux through the electron transport chain contributes to metabolic stress. The rescue of cells with thiourea and 2,2'-bipyridine during Levo treatment suggests that Levo killing is mediated, in part, by the accumulation of reactive metabolic species (52, 53).

While *P. aeruginosa* maintains reducing activity late in stationary phase, this reducing activity appears insufficient to sustain ATP concentrations at exponential-phase levels (Fig. S10). Therefore, we asked whether macromolecular synthesis could explain the low ATP abundance in these cells.

## Metabolic activity in the stationary phase drives transcription

We hypothesized that the sustained reductase activity of stationary-phase *P. aeruginosa* generates ATP that is consumed during nucleic acid synthesis. Transcription requires active topoisomerases to relieve topological stress and could render stationary-phase *P. aeruginosa* more sensitive to fluoroquinolones than non-transcribing cells (55). We measured the nucleic acid synthesis of 16 h cultures by the incorporation of tritiated

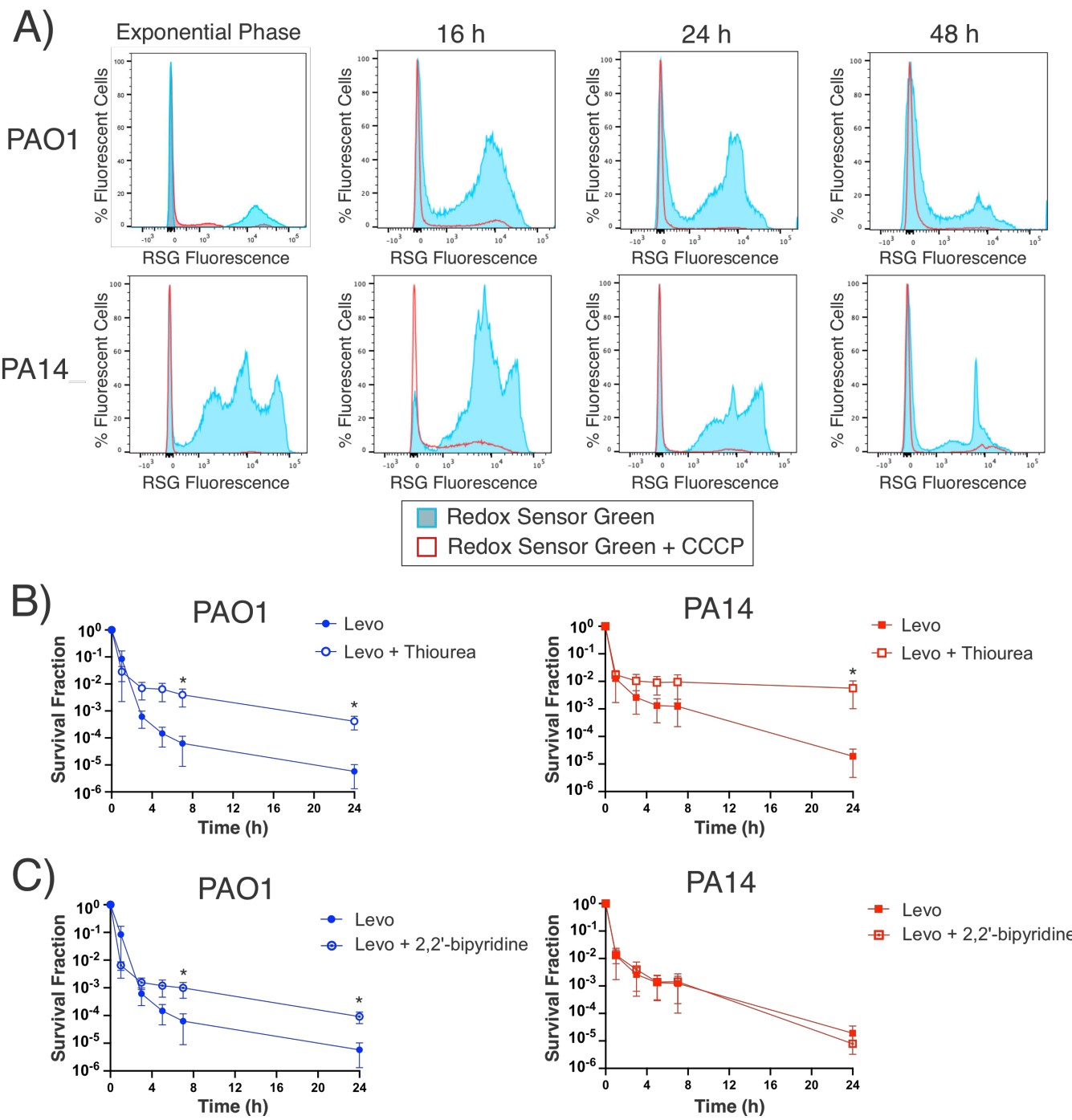

**FIG 5** *P. aeruginosa* sustains reductase activity in the stationary phase and co-treatment with antioxidants can partially rescue Levo-mediated killing. (A) Reductase activities were measured by adding Redox Sensor Green (RSG) to cultures grown for the specified times and then monitoring fluorescence by flow cytometry. Histograms are representative of three biological replicates. (B) During Levo treatment, the addition of the antioxidant thiourea (150 mM) decreased cell death. (C) Addition of an iron chelator (0.3 mM 2,2′-bipyridine) during Levo treatment decreased cell death in PAO1 cultures, but not PA14. Data are representative of at least three biological replicates and error bars represent the standard error of the mean. Asterisks denote statistical significance ($P \leq 0.05$ by Mann-Whitney test with multiple test correction).

uridine into *de novo* nucleic acids (Fig. 6A). As expected, stationary-phase *E. coli* MG1655 has minimal radiolabeled uridine in its extracted nucleic acids, but stationary-phase *P. aeruginosa* PAO1 and PA14 both have appreciable nucleic acid synthesis activity (56). Degradation of alkaline-labile RNA in each sample with 3 M KOH before scintillation

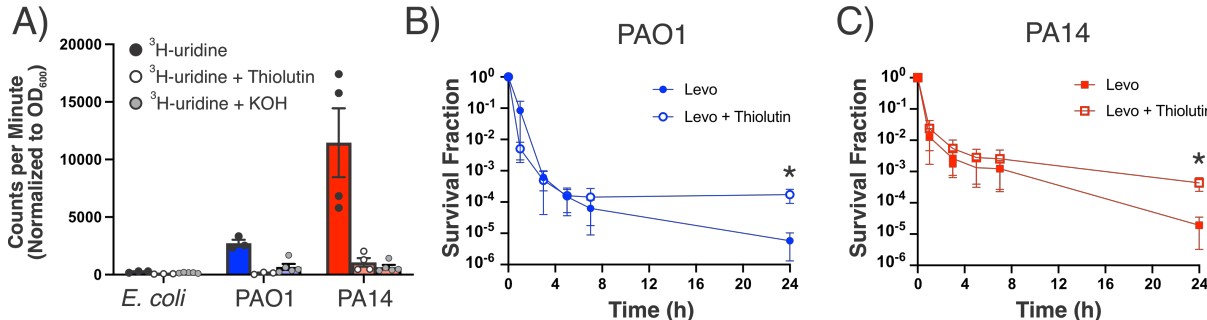

**FIG 6** Stationary-phase *P. aeruginosa* maintains transcription and can be partially rescued from Levo-mediated killing by addition of a transcription inhibitor. (A) Nucleic acid synthesis was measured in *E. coli* MG1655 and *P. aeruginosa* PAO1 and PA14 by adding tritiated uridine to 16 h cultures and measuring the radioactivity of nucleic acid fractions (by scintillation of trichloracetic acid cell extracts) after 1 h incubation. Treatment with the RNA polymerase inhibitor thiolutin (100 µg/mL for *P. aeruginosa*, 8 µg/mL for *E. coli*) or with 3 M KOH significantly reduced uridine incorporation ($P \leq 0.05$, Kruskal-Wallis test). Radiolabeled uridine incorporation is reported as counts per minute normalized to the $OD_{600}$ of the specific culture. Thirty minutes pre-treatment with 100 µg/mL thiolutin increase the survival of 16 h *P. aeruginosa* (B) PAO1 and (C) PA14 cultures against Levo. Data represent at least three biological replicates and error bars represent the standard error of the mean.

counting reduced the counts of tritiated uridine. Consistent with these findings, the addition of an RNA polymerase inhibitor, thiolutin, along with tritiated uridine drastically decreased the amount of newly synthesized nucleic acids. These data suggest that the nucleic acid synthesis activity in stationary-phase cultures is primarily due to RNA, not DNA, synthesis.

Because bacterial transcription is tightly coupled to translation, we sought to determine whether protein synthesis was also maintained in stationary-phase *P. aeruginosa* cultures. By measuring cells' incorporation of an azidated methionine analog [azidohomoalanine (AHA)] that could be conjugated to an alkylated fluorophore using click chemistry, we found that stationary-phase cultures of *P. aeruginosa* had significantly fewer cells that were actively translating and significantly lower levels of translation compared to exponential-phase cultures (Fig. S11). These data suggest that the macromolecular synthesis of stationary-phase *P. aeruginosa* can be attributed to nucleic acid synthesis—specifically transcription—and not protein synthesis.

If active transcription confers Levo sensitivity, we would expect transcription inhibition to rescue Levo-treated *P. aeruginosa*. When we paused transcription with thiolutin for 30 min before Levo treatment, we found that *P. aeruginosa* survival was significantly increased by the end of treatment (24 h) compared to cells that did not receive thiolutin (Fig. 6B and C). However, thiolutin did not completely rescue the cells. In light of the incomplete rescuing effects of thiourea and 2,2′-bipyridine, we reason that Levo killing can be partially attributed to the levels of cellular transcription and also to the oxidative stress that cells incur.

## DISCUSSION

In this study, we delved into the responses of stationary-phase *P. aeruginosa* to Levo treatment with the two most common reference strains: PAO1 and PA14 (35). In contrast to previous findings in *E. coli* fluoroquinolone persisters, we discovered that *P. aeruginosa* cells retain redox and transcriptional activities in the stationary phase and succumb to Levo during treatment (25, 48, 56). While the nature of clinical infections is far more complex than laboratory monocultures, our findings are nonetheless relevant to understanding how genetically susceptible *P. aeruginosa* cells survive antibiotic treatment. A recent study of chronic wound infections of *P. aeruginosa* in mice suggests that, beyond the expected organization of cells as biofilms or aggregates, single cells are distributed planktonically throughout wounds, lending credence to the study of planktonic cultures (57). Furthermore, our use of carbon-depleted cultures contributes an informative perspective on *P. aeruginosa* antibiotic susceptibility. Most bacteria

exist in relative quiescence in naturally nutrient-limited environments, yet antibiotic susceptibility has traditionally been studied in the context of growing cells or nutrient-rich media (20, 21, 33). In future studies, we will determine how well our findings translate to alternative models of *P. aeruginosa* infection *in vitro* and *in vivo*.

We found that transcription is active in stationary-phase *P. aeruginosa* cultures that exhibit minimal translational activity. This begs the question: what are cells transcribing if not mRNA for protein translation? One hypothesis is that the measured transcription levels are a result of pervasive transcription that allows cells to surveil the genome for lesions (58). Alternatively, it may be the expression of small RNAs that allow for rapid cellular responses during periods of slow growth when translation of a protein-mediated response would be energetically costly and slow (59). Many sRNAs, rRNAs, transcriptional regulators, and elongation factors are upregulated in stationary-phase *P. aeruginosa*, supporting the hypothesis that continued expression of regulatory genes is central to stationary-phase homeostasis (22, 60). Elucidating the repertoire of genes that *P. aeruginosa* expresses during growth arrest would bridge our knowledge gap of how this pathogen survives in infected hosts or natural habitats.

Along this vein, the killing kinetics of *P. aeruginosa* treated with Levo and thiolutin show that the population's increased survival after 24 h is attributed to a subpopulation of cells that are rescued from death between 7 and 24 h (Fig. 6B and C). This may suggest that, beyond the population's basal transcription level, subpopulations of *P. aeruginosa* induce transcriptional responses, such as the stringent response, that prove fatal later in Levo treatment (10, 61). Thiolutin thus inhibits these responses and rescues cells. In light of this finding, the combination of Levo with drugs that inhibit transcription or relieve redox stress should be avoided, as they could limit fluoroquinolone efficacy (as shown in Fig. 5B and C). Future work should seek to determine whether subpopulations that survive Levo activate different transcriptional programs or simply remain transcriptionally quiescent until after treatment ceases.

The reduced translational activity in stationary-phase *P. aeruginosa* compared with their exponentially growing counterparts may explain why 16 h cultures of PAO1 and PA14 are tolerant to ribosome-inhibiting Tobramycin. Alternatively, because aminoglycoside uptake is proton-motive force dependent, one might hypothesize that stationary-phase *P. aeruginosa* has low proton motive force and limited Tobramycin transport (62). However, we reason that our stationary-phase *P. aeruginosa* cultures maintain their proton motive force because reductase activity—as measured via RSG fluorescence—was decreased by CCCP, which uncouples the proton gradient. Additionally, the vast majority of cells do not have compromised membranes, as evidenced by a lack of SytoxBlue staining of stationary-phase cultures (Fig. S4).

It is worthwhile to note that the proportions of surviving cells at the end of Levo treatment are higher for our microscopy experiments than the overall survival levels after treatment in test tubes in the drug concentration-dependent killing assays. There are several possibilities for these discrepancies. First, cells can die after the antibiotic is removed during the post-treatment recovery phase, as has been reported for *E. coli* (37, 63). Survivors in the drug assays are enumerated by colony-forming units on nutrient-rich agar media after treatment, whereas for microscopy experiments, the number of non-viable cells was quantified without a recovery period during which more cells would be expected to die. Second, the cells that remained PI-negative and visibly intact under the microscope may have been viable but non-culturable and thus would overestimate the proportion of cells that are Levo persisters. In other words, microscopy image quantification portrays cell viability at the end of treatment, but does not capture a cell's ability to survive and resume growth after treatment that would qualify it as a persister. Finally, we acknowledge the possibility that cell survival may be different in test tubes as opposed to on agarose pads under the microscope. The use of a microfluidic chamber that can exchange between media containing Levo and fresh, antibiotic-free media for recovery would help determine the fidelity of the microscopy experiments to planktonic cultures.

This study raises key differences between *P. aeruginosa* phenotypic responses to fluoroquinolone antibiotics compared to *E. coli*. According to previous observations, *E. coli* cells only lyse during the post-treatment recovery period (37, 64). It is thought that death occurs after fluoroquinolone removal because nongrowing cells do not face the consequences of trapped topoisomerases on their DNA until DNA synthesis or transcription resumes (47, 48). Sustained transcription in stationary-phase *P. aeruginosa* could therefore release trapped topoisomerases and lead to broken DNA ends during Levo treatment. We hypothesize that, like *E. coli*, *P. aeruginosa* persisters also require DNA repair to survive Levo treatment. However, because *P. aeruginosa* Δ*recA* strains had similar survival against Levo as wild-type cultures, we propose that this repair does not rely on cells' ability to activate the SOS response. To our knowledge, this is the first demonstration that RecA, a hallmark of fluoroquinolone persistence in *E. coli*, is not a major contributor to *P. aeruginosa* fluoroquinolone persistence. Investigating *P. aeruginosa*-specific mechanisms of tolerance and persistence will better inform therapeutic strategies to treat chronic and recurrent infections of antibiotic-susceptible isolates.

## MATERIALS AND METHODS

### Bacterial strains and strain construction

The *P. aeruginosa* and *E. coli* strains that were used in this study are listed in Table S2. Knockout strains of *P. aeruginosa* were generated by allelic exchange and verified by colony PCR using primers listed in Table S3 (65). One clone of each mutant was also verified by WGS (Fig. S6). Detailed methods are included in the supplemental materials.

### Culture media and antibiotics

All chemicals and reagents were purchased from MilliporeSigma or Fisher Scientific unless otherwise stated. Cells were inoculated from frozen stocks into nutrient-rich media: cation-adjusted Mueller-Hinton Broth (CA-MHB) for *P. aeruginosa* or lysogeny broth (LB) for *E. coli*. For assays, inocula were diluted into chemically defined Basal Salt Media (BSM) with 15 mM succinate for *P. aeruginosa* and Gutnick media with 10 mM glucose (Gutnick-glucose) for *E. coli* (66–68). Media, antibiotic, and chemical stock preparations are detailed in the supplemental materials.

### Growth curves

*P. aeruginosa* strains were inoculated into test tubes (16 mm diameter) containing 2 mL CA-MHB and grown at 37°C, shaking at 250 rpm. After 4–5 h of growth, inocula were diluted 100-fold into 2 mL BSM in test tubes. After overnight growth (16 h), cells were diluted to an optical density at 600 nm ($OD_{600}$) of 0.001–0.005 in a 250 mL baffled Erlenmeyer flask with 25 mL BSM and cultured at 37°C. $OD_{600}$ readings were measured hourly on a Biotek multimode plate reader for 12 h and again at 16 and 28 h. For *E. coli* growth curves, cells were inoculated in LB and diluted 200-fold into Gutnick-glucose media for overnight growth. After overnight growth (16 h), *E. coli* was diluted to $OD_{600}$ <0.01 in flasks of 25 mL Gutnick-glucose media. For growth curves with SytoxBlue (ThermoFisher S11348), SytoxBlue dye was added to media for a final concentration of 2.5 µM.

### Minimum inhibitory concentration assays

Antibiotic minimum inhibitory concentrations (MICs) were determined by using MIC test strips from Liofilchem (Fig. S1). The MICs of thiolutin, thiourea, and 2,2′-bipyridine were determined by broth microdilution method following CLSI standards (69). Details are included in the supplemental material.

## Antibiotic persistence assays

*P. aeruginosa* strains were inoculated into 2 mL CA-MHB and grown at 37°C, shaking at 250 rpm (Fig. S1). After 4–5 h of pre-growth, inocula were diluted 1:100 into 250 mL baffled flasks with 25 mL BSM. Following growth to stationary phase (16, 24, or 48 h, as indicated), $OD_{600}$ of each culture was measured and 10 µL of cells was collected for serial dilution in PBS and plating onto CA-MHB agar plates for colony-forming unit (CFU) enumeration. Then, the culture was aliquoted into 2 mL aliquots in test tubes and treated with 10 µL antibiotic for the final concentrations as listed in each figure. To elucidate the effects of antioxidants on Levo persistence of *P. aeruginosa*, bacteria were treated with 0.3 mM of 2,2′-bipyridine or 150 mM of thiourea together with 5 µg/mL of Levo. To determine how transcription inhibition affects Levo persistence, 100 µg/mL thiolutin (1× MIC) was added 30 min before Levo administration. Thiolutin was used in lieu of the more common transcriptional inhibitor, Rifampicin, because PA14 is intrinsically Rifampicin-resistant.

At designated time points during treatment, 500 µL of culture was collected and pelleted by centrifugation at 21,000 rcf for 3 min. After removing 450 µL of supernatant, the pellets were washed with 450 µL PBS. This step was repeated, effectively diluting the antibiotics to subinhibitory levels (100-fold dilution). The cells were then serially diluted in PBS and 10 µL spots of each dilution were plated onto CA-MHB agar for CFU enumeration.

*E. coli* was inoculated from frozen stocks into 2 mL LB and diluted 200-fold into 25 mL Gutnick-glucose. Cultures were treated and sampled as described for *P. aeruginosa*, with serially diluted cells plated to LB agar plates instead of CA-MHB agar.

## Time-lapse fluorescence microscopy experiments

Details on sample preparation, image acquisition, and image analysis are included in the supplemental materials (Fig. S1). In brief, cells were cultured overnight in chemically defined media (BSM for *P. aeruginosa* or Gutnick-glucose for *E. coli*) with 2.5 µM SytoxBlue. 1.5% (wt/vol) agarose pads containing 5 µg/mL Levo and 16 µM propidium iodide (Invitrogen L34856) were prepared in a Bioptechs Interchangeable Coverglass Dish using cell-free conditioned media from matched overnight cultures that were grown in chemically defined media *without* SytoxBlue. Cells were diluted 30-fold in PBS before seeding to solidified agarose pads. Samples were kept at 37°C during imaging in a PeCon live cell incubation chamber with a humidifying lid. Samples were imaged on a customized Zeiss Axiovert 200M microscope with a Plan-Apochromat 63×/1.40 Oil Ph3 M27 objective using the software MetaMorph Premier version 7.10.5 (Molecular Devices).

## Metabolic assays

The BacLight RSG assay was conducted according to the manufacturer's protocol (Invitrogen B34954). In brief, the $OD_{600}$ of cultures at designated times (exponential phase, 16, 24, or 48 h) were measured and cells were diluted to an $OD_{600}$ of ~0.01 in 1 mL PBS in flow cytometry tubes. As a positive control for stationary-phase *E. coli* MG1655, cells were treated with 10 mM glucose to stimulate reductase activity. The diluted cultures and negative controls in which bacteria were treated with 2.5 µM CCCP (2 µL Component D) were incubated statically at 37°C for 5 min in the dark. Then, all tubes received 1 µM RSG (1 µL Component A) and were incubated statically at 37°C for 10 min in the dark. Samples were analyzed on an LSRII flow cytometer with FACS DiVa (BD Biosciences). Fluorescence from each cell was detected using the green fluorescence filter (525/50 nm band-pass). Data were analyzed using FlowJo (BD Biosciences).

Details on the additional assays used to characterize stationary-phase cultures (ATP concentrations by BactiterGlo assay, succinate concentrations by fluorometric assay, and translation by Click-IT AHA assay) are included in the supplemental materials.

## Quantification of nucleic acid synthesis

The incorporation of [5,6-$^3$H]-uridine [PerkinElmer (now Revvity), Hopkinton, MA] into newly synthesized nucleic acids was measured in stationary-phase *P. aeruginosa* and *E. coli*. In brief, aliquots of each culture were incubated for 1 h with 1 µCi of tritiated uridine with or without thiolutin at approximately the MIC for each species (100 µg/mL for *P. aeruginosa* or 8 µg/mL for *E. coli*). Nucleic acids were extracted in ice-cold 10% trichloroacetic acid (TCA) and incubated on ice for 30 min. For KOH-treated samples, 0.1 vol of 3 M KOH was added after incubation with uridine and samples were incubated statically at 37°C for 24 h to hydrolyze labile RNA; then, nucleic acids were extracted with 10% TCA for 30 min and processed the same as other samples. The precipitates from each sample were collected by vacuum filtration onto 25 mm Whatman GF/C glass filters and washing three times each with excess ice-cold 10% TCA and ice-cold 70% ethanol. Filters were air dried then transferred into scintillation vials with EconoSafe scintillation fluid. Radioactivity was measured using a Beckman-Coulter LS6500 liquid scintillation counter. Counts per minute (CPM) from the negative control (an unused Whatman GF/C glass filter directly transferred to scintillation fluid) were subtracted from each sample's CPM and normalized to the $OD_{600}$ of each sample.

## Statistical analyses

At least three biological replicates were performed for each experiment unless otherwise noted. Plotting and analysis were conducted using GraphPad Prism. Specific statistical tests are detailed in the figure captions.

## ACKNOWLEDGMENTS

The authors thank Dr. Keith Poole for distributing *P. aeruginosa* PAO1 (K767), and Dr. Mona Wu Orr for distributing *E. coli* HB101 pRK2013 and *E. coli* DH5α pEX18Gm. *P. aeruginosa* PA14 and bacteriophages JBD90 and JBD23 were obtained through BEI Resources, NIAID, NIH. The authors thank the following people at UConn Health for their assistance in conducting experiments: Ms. Susan Staurovsky (Center for Cell Analysis and Modeling Microscopy Facility), Dr. Evan Jellison and Ms. Li Zhu (Flow Cytometry Core), Mr. Kevin Higgins and Mr. Rob Speers (Office of Radiation Safety), Dr. Bo Reese (Center for Genome Innovation), and Dr. Vijender Singh (Computational Biology Core). The authors are also grateful to Dr. Christina Stalling and Ms. Helen Blaine of Washington University for sharing their nucleic acid synthesis protocol. Finally, the authors thank Dr. Peter Setlow for his thoughtful feedback on our manuscript.

This work was supported by funding awarded to W.K.M. from the University of Connecticut start-up fund and the National Institutes of Health (NIH; DP2GM146456-01). P.J.H. is additionally supported by the National Institutes of Health (F30DE032598) and was previously supported by the NIH Skeletal, Craniofacial, and Oral Biology Training Grant (T90DE021989-11). Y.I.W. is supported by the National Institutes of Health (NIH; GM117061). The funders had no role in the design of our experiments or preparation of this manuscript.

## AUTHOR AFFILIATIONS

[1]Department of Molecular Biology & Biophysics, UConn Health, Farmington, Connecticut, USA
[2]School of Dental Medicine, UConn Health, Farmington, Connecticut, USA
[3]Richard D. Berlin Center for Cell Analysis and Modeling, UConn Health, Farmington, Connecticut, USA

## AUTHOR ORCIDs

Patricia J. Hare  http://orcid.org/0000-0001-5014-6161

Wendy W. K. Mok  http://orcid.org/0000-0002-6638-2959

## FUNDING

| Funder | Grant(s) | Author(s) |
|---|---|---|
| HHS | National Institutes of Health (NIH) | DP2GM146456-01 | Wendy W. K. Mok |
| HHS | NIH | National Institute of Dental and Craniofacial Research (NIDCR) | F30DE032598, T90DE021989-11 | Patricia J. Hare |
| HHS | National Institutes of Health (NIH) | GM117061 | Yi I. Wu |

## AUTHOR CONTRIBUTIONS

Patricia J. Hare, Conceptualization, Data curation, Formal analysis, Funding acquisition, Investigation, Methodology, Supervision, Validation, Visualization, Writing – original draft, Writing – review and editing | Juliet R. Gonzalez, Data curation, Writing – review and editing | Ryan M. Quelle, Data curation, Writing – review and editing | Yi I. Wu, Methodology, Resources, Writing – review and editing | Wendy W. K. Mok, Conceptualization, Data curation, Formal analysis, Funding acquisition, Investigation, Methodology, Resources, Supervision, Validation, Visualization, Writing – original draft, Writing – review and editing

## DATA AVAILABILITY

Whole-genome sequence reads from PAO1 Δ*recA* and PA14 Δ*recA* can be found in the NCBI Sequence Read Archive under BioProject PRJNA1034140.

## ADDITIONAL FILES

The following material is available online.

### Supplemental Material

**Supplemental material (Spectrum03567-23-s0001.docx).** Supplemental methods, Tables S1 to S4, Figures S1 to S11, and captions for Videos S1 to S9.
**Video S1 (Spectrum03567-23-s0002.avi).** Wild-type PAO1 during Levo tx_replicate 1.
**Video S2 (Spectrum03567-23-s0003.avi).** Wild-type PAO1 during Levo tx_replicate 2.
**Video S3 (Spectrum03567-23-s0004.avi).** Wild-type PA14 during Levo tx_replicate 1.
**Video S4 (Spectrum03567-23-s0005.avi).** Wild-type PA14 during Levo tx_replicate 2.
**Video S5 (Spectrum03567-23-s0006.mov).** *E. coli* MG1655 during Levo treatment_replicates 1 and 2.
**Video S6 (Spectrum03567-23-s0007.avi).** PAO1 *recA* knockout during Levo tx_replicate 1.
**Video S7 (Spectrum03567-23-s0008.avi).** PAO1 *recA* knockout during Levo tx_replicate 2.
**Video S8 (Spectrum03567-23-s0009.avi).** PA14 *recA* knockout during Levo tx_replicate 1.
**Video S9 (Spectrum03567-23-s0010.avi).** PA14 *recA* knockout during Levo tx_replicate 2.

### Open Peer Review

**PEER REVIEW HISTORY (review-history.pdf).** An accounting of the reviewer comments and feedback.

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
