## [Reviewer comments · Microbiology Spectrum]

Microbiology Spectrum

Metabolic and Transcriptional Activities Underlie Stationary-Phase *Pseudomonas aeruginosa* Sensitivity to Levofloxacin

Patricia Hare, Juliet Gonzalez, Ryan Quelle, Yi Wu, and Wendy Mok

Corresponding Author(s): Wendy Mok, UConn Health

Review Timeline:

Submission Date:	October 5, 2023
Editorial Decision:	October 24, 2023
Revision Received:	November 6, 2023
Accepted:	November 16, 2023

Editor: Minsu Kim

Reviewer(s): Disclosure of reviewer identity is with reference to reviewer comments included in decision letter(s). The following individuals involved in review of your submission have agreed to reveal their identity: Gregory Wiedman (Reviewer #1)

Transaction Report:

DOI: <https://doi.org/10.1128/spectrum.03567-23>

Re: Spectrum03567-23 (Metabolic and Transcriptional Activities Underlie Stationary-Phase *Pseudomonas aeruginosa* Sensitivity to Levofloxacin)

Dear Dr. Wendy W. K. Mok:

Thank you for sending the manuscript to us. The manuscript was reviewed by two reviewers. Both reviewers found the work interesting, and the response was positive. Below you will find their comments as well as instructions from the Spectrum editorial office. The second reviewer directly commented on the manuscript (attached).

Please address their comments and return the manuscript within 60 days; if you cannot complete the modification within this time period, please contact me. If you do not wish to modify the manuscript and prefer to submit it to another journal, notify me immediately so that the manuscript may be formally withdrawn from consideration by Spectrum.

Revision Guidelines

Sincerely,
Minsu Kim
Editor
Microbiology Spectrum

Reviewer #1 (Comments for the Author):

The reviewer would like to thank the authors for their submission which is titled: "Metabolic and Transcriptional Activities Underlie Stationary-Phase *Pseudomonas aeruginosa* Sensitivity to Levofloxacin". In this work, the authors hypothesize that the apparent sensitivity of stationary-phase *P. aeruginosa* to Levofloxacin is mediated by the active metabolism of the cell and does not involve SOS response. They address this hypothesis by examining bacterial cell morphology and metabolism in the presence of Levo and making inferences from such data. The reviewer would like to provide a critique of the article on the basis

of the stylistic aspects (major and minor) as well as the technical aspects (major and minor)

Stylistic (minor)

1. If the authors could briefly clarify for the readers in the caption that the breakpoints are *not* conducted with stationary phase bacteria this might be best. The author understands what they intended to convey but at a cursory glance readers may mistake their data in red and blue for depicting resistance in a standard MIC assay.

Stylistic (major)

No Major Stylistic Issues

Technical (minor)

1. In Figure 5A, the data representing PA14, there appear to be multiple peaks for RGS fluorescence in their flow data. Do the authors believe these are significant? Does RGS uptake and activity with respect to reductases generally exhibit an all-or-none response or are there variations in their experience?

2. The authors seem to suggest that there is no difference in survival among the wild-type and Δ recA mutants. Examining Figure 4C, specifically PAO1 vs PAO1 Δ recA, is this true for all concentrations? The reviewer would draw attention to the apparent 0.5 μ g/mL data point.

3. As a follow-up to Major Technical Point 2, the specific line in the text reads, "stationary-phase *P. aeruginosa* recA knockouts and their wildtype counterparts across multiple Levo concentrations". Is the reviewer to interpret that the phrase "multiple" is used in place of "all" due to the point discussed above?

Technical (major)

No Major Technical issues

Reviewer #2 (Comments for the Author):

The authors have done extensive work on pseudomonas isolates with the aim of elucidating the mechanisms behind levofloxacin tolerance.

The experiments are well-designed and executed.

However, while the text reads well, the lack of tangible data made it difficult to read, particularly for someone who is coming from a clinical bacteriology background.

An overall figure that explains the different experiments done (a conceptual framework if I may say so) with a hierarchical flow chart would make this work more easier to comprehend.

Other comments are attached to the manuscript

Editor's Comments:

Thank you for sending the manuscript to us. The manuscript was reviewed by two reviewers. Both reviewers found the work interesting, and the response was positive. Below you will find their comments as well as instructions from the Spectrum editorial office. The second reviewer directly commented on the manuscript (attached).

Please address their comments and return the manuscript within 60 days; if you cannot complete the modification within this time period, please contact me. If you do not wish to modify the manuscript and prefer to submit it to another journal, notify me immediately so that the manuscript may be formally withdrawn from consideration by Spectrum.

Thank you for considering our manuscript and for the opportunity to revise our work according to the reviewers' feedback.

Reviewer 1:

The reviewer would like to thank the authors for their submission which is titled: "Metabolic and Transcriptional Activities Underlie Stationary-Phase *Pseudomonas aeruginosa* Sensitivity to Levofloxacin". In this work, the authors hypothesize that the apparent sensitivity of stationary-phase *P. aeruginosa* to Levofloxacin is mediated by the active metabolism of the cell and does not involve SOS response. They address this hypothesis by examining bacterial cell morphology and metabolism in the presence of Levo and making inferences from such data. The reviewer would like to provide a critique of the article on the basis of the stylistic aspects (major and minor) as well as the technical aspects (major and minor)

We would like to thank Reviewer 1 for their suggestions on how to improve our manuscript. Please find our responses to each item below.

Stylistic (minor)

1. If the authors could briefly clarify for the readers in the caption that the breakpoints are *not* conducted with stationary phase bacteria this might be best. The author understands what they intended to convey but at a cursory glance readers may mistake their data in red and blue for depicting resistance in a standard MIC assay.

Thank you for this recommendation. We have now clarified that MIC assays are not conducted on stationary-phase bacteria and have adjusted the caption of Fig. 1 to read, "Minimum inhibitory concentrations of *P. aeruginosa* PAO1 and PA14, as measured by testing growth inhibition of low-density cultures with MIC test strips (n>3)."

Stylistic (major)

No Major Stylistic Issues

Technical (minor)

1. In Figure 5A, the data representing PA14, there appear to be multiple peaks for RGS fluorescence in their flow data. Do the authors believe these are significant? Does RGS uptake and activity with respect to reductases generally exhibit an all-or-none response or are there variations in their experience?

Our histograms of RSG fluorescence, like those from other publications (PMID: 29046671, 30332894), typically show a range of fluorescence intensities. Although the multiple peaks we

observe are somewhat unusual, we do not know if they are biologically significant and will have to investigate further in future experiments.

2. The authors seem to suggest that there is no difference in survival among the wild-type and $\Delta recA$ mutants. Examining Figure 4C, specifically PAO1 vs PAO1 $\Delta recA$, is this true for all concentrations? The reviewer would draw attention to the apparent 0.5 $\mu\text{g}/\text{mL}$ data point.

We appreciate the reviewer's careful review of the data in Fig. 3C. To determine if there were significant differences in survival between the PAO1 wildtype and $\Delta recA$ at any of the Levo concentrations, we repeated our statistical analysis and confirmed that the differences in survival are not statistically significant ($p > 0.05$ by Mann-Whitney test with multiple test correction). The caption for Figure 3C has been updated to include this analysis.

3. As a follow-up to Major Technical Point 2, the specific line in the text reads, "stationary-phase *P. aeruginosa* *recA* knockouts and their wildtype counterparts across multiple Levo concentrations". Is the reviewer to interpret that the phrase "multiple" is used in place of "all" due to the point discussed above?

The $\Delta recA$ strain's Levo persistence is not different from the wildtype at any of the concentrations that we tested, but we did not want to use the word "all" because we did not exhaustively test all Levo concentrations from zero drug to complete killing. To make it clearer, we have adjusted the text from "...and their wildtype counterparts across multiple Levo concentrations," to "...and their wildtype counterparts across the range of Levo concentrations that we tested."

Technical (major)

No Major Technical issues

Reviewer 2:

The authors have done extensive work on pseudomonas isolates with the aim of elucidating the mechanisms behind levofloxacin tolerance.

The experiments are well-designed and executed.

However, while the text reads well, the lack of tangible data made it difficult to read, particularly for someone who is coming from a clinical bacteriology background. An overall figure that explains the different experiments done (a conceptual framework if I may say so) with a hierarchical flow chart would make this work more easier to comprehend. Other comments are attached to the manuscript

We would like to thank Reviewer 2 for their suggestions on how to improve our manuscript, especially in regards to improving accessibility for more clinical audiences.

We have now included a supplemental figure that provides an overview of the key experiments we conducted in the manuscript to anchor readers (Fig. S1).

Main Text:

"Antibiotic Recalcitrant": May be better to use a term that is more commonly used in describing bacterial characteristics, as this term is specific to biofilms, to the best of my knowledge and it seems like the authors are implying general resistance than something specific to biofilms

To clarify our text as the reviewer suggests, we have edited the wording to “antibiotic-refractory,” which encompasses antibiotic treatment failure by resistance, tolerance, or persistence (PMID 20980069).

Is a biofilm considered as a reversible phenotypic change or a different growth pattern?
Reconsider the wording please

We consider a biofilm to be an example of a reversible phenotypic growth pattern because individual cells can reversibly express gene programs for attachment or motility/dispersal. We have tried to improve the sentence's phrasing to focus more on individual cells within biofilms than the biofilm itself:

Changed from “...by undergoing reversible phenotypic changes, like forming biofilms” to “...by undergoing reversible phenotypic changes, as observed for bacteria in biofilms.”

“Stochastic”: Consider using a term that is more commonly used in bacteriology

We find the term “stochastic” is commonly used to describe the random/chance variations among single cells in isogenic populations in the field of persister biology. For example, one of the leaders in the field, Kim Lewis, published a paper in *PLOS Biology* in 2021 entitled, “Bacterial persisters are a stochastically formed subpopulation of low-energy cells.” Other notable examples include PMIDs 27980159 and 19029898.

“Perturbations”: May be better to use a term that is more commonly used in describing bacterial characteristics

We have re-written the sentence for clarity as the reviewer suggests: “In *P. aeruginosa*, metabolic heterogeneity due to differential nutrition and oxygenation has been shown to impact antibiotic tolerance.”

Lines 98-98: This para is a too lengthy. Better restrict to the key finding or two only, to avoid multiple duplications

We have removed sentences in this paragraph to shorten it and highlight the most important findings as recommended.

Line 102: While the title states stationary phase, the results does not state anything about the phase. There needs to be a better description indicating that this results are pertinent to the stationary phase experiments.

We specified the phase of growth in the following places in the paragraph: “We began by asking whether the persistence of stationary-phase cultures of *P. aeruginosa*” (Lines 105-106) and “we performed survival assays at supra-MIC concentrations to determine how sensitive stationary-phase PAO1 and PA14 were to each antibiotic” (Lines 111-113).

“Highly sensitive”: Too subjective

Thank you for pointing this out. We have removed the word “highly” to more objectively state that stationary-phase *P. aeruginosa* is sensitive to Levo, but not the other antibiotic compounds.

Line 102: Was this experiment done for *E. coli*, if not why?

This experiment was not done with *E. coli* because the focus of our investigation was on *P. aeruginosa*. *E. coli* was used in later experiments as a point of comparison against which we could highlight Levo-treated *P. aeruginosa* phenotypes.

Line 107: Differ in what? In response against antibiotics?

To enhance the clarity as the reviewer suggests, we have broken the sentence into three parts: “We began by asking whether the persistence of stationary-phase cultures of *P. aeruginosa* would differ against bactericidal antibiotics with different primary targets. For generalizability, we tested *P. aeruginosa* strains PAO1 and PA14—representing the two major genetic clades of this species—in each experiment. ... Stationary-phase cultures were grown in chemically defined media with succinate as the sole carbon source before treatment.”

Line 111: Pls give the values of the MIC, it is difficult to make it out from the figure alone.

As the reviewer suggested, we have added a new table to the supplement (Table S2) with the MIC values for Levo, Aztreonam, and Tobramycin so that the values can be elucidated more easily.

Line 114: Why were these multiplications used?

The aztreonam MIC is approximately 4 ug/ml, according to the figure 1 they were tested in to 50 only. One has to then refer to the supplementary figures to find the data for this. This makes the paper difficult to read and comprehend. Remove figure S1 and include it within figure 1 itself where the complete concentration range tested are included in 1B,C and D. Also, give the exact reductions noted in Tobramycin in the figure

As suggested, we now include the full datasets for Aztreonam and Tobramycin in the main figure panel 1A.

Line 115: Write in a quantitative manner. This is a subjective statement

We have re-written the statement to be more objective and quantitative:

“We found that *P. aeruginosa* was completely tolerant to Tobramycin at 15X MIC and Aztreonam up to 25-50X MIC (Fig. S1). Comparatively, *P. aeruginosa* was highly sensitive to Levo; we detected approximately six-log decreases in cell survival when Levo was administered at 15X MIC (5 µg/mL).”

Line 116: State the actual decrease.

We now state the decreased survival as “six-log” instead of “greater than five-log.”

Line 117: At stationary phase?

We have changed the title to “Stationary-phase *P. aeruginosa* dies during Levo treatment” for clarity.

Lines 122-124: The results given this way without any data is not very convincing

Was SytoxBlue completely removed before the cells were inoculated on the agarose pads? May need a statement here

To clarify this section as suggested, the sentences in this paragraph have been rearranged to emphasize that the cells from SytoxBlue-containing cultures were diluted 30-fold in PBS before being seeded on the agarose pad for imaging, thus SytoxBlue is nearly absent from these samples. The data for SytoxBlue are shown in Fig. S4 in the Supplemental Materials and show that few cells are SytoxBlue-positive.

Line 126: What was the concentration of Levo used here? And what folds of the MIC? Why was SytoxBlue used in the earlier phase and PI here?

As the reviewer suggests, the Levo concentration is now explicitly stated in the paragraph.

To enhance our clarity, the paragraph has been re-written to emphasize the purpose of switching from SytoxBlue during growth to stationary phase and PI during treatment:

“Any cells that were SytoxBlue-positive were classified as having died during the growth to stationary phase, before seeding onto the agarose pad, whereas PI+ cells were classified as having died after exposure to Levo in the agarose pad.”

Lines 127-128: Need to mention if the same was done for *E. coli*

The data and caption in Fig. S2 clarify that the control experiments were also done for *E. coli*.

Lines 129-131: Why were these not repeated with aztreonam and tobramycin?

Because the unusual phenotype (several log-fold decrease in survival of stationary-phase cultures at 15X MIC) was only observed for Levo and not Aztreonam or Tobramycin, we focused the rest of our investigation on understanding Levo tolerance/susceptibility. This is why we specifically conducted time-lapse microscopy experiments for cells treated with Levo. Additionally, performing the time-lapse imaging experiments at the Core Facility with all three drugs would be prohibitively expensive and time-consuming.

Lines 132-133: What was the concentration used and what folds of the MICs

For clarity, we state in the caption of Fig. S3 that Levo was used in the same concentration as for *P. aeruginosa* (5 µg/mL). The Levo MIC of *E. coli* is 0.03 µg/mL (PMID: 34097492), so 5 µg/mL Levo is 150X MIC for *E. coli* and 15X MIC for *P. aeruginosa*. This further emphasizes how unexpected *P. aeruginosa* Levo sensitivity was since, at only 15X MIC, cells display an active response to the antibiotic that *E. coli* cells do not demonstrate even at 150X MIC.

Lines 136-140: How confident are the authors of this statement without actually looking for culturability? What are the specificities and sensitivities of the two dyes used? Can the stress ensured during processing contribute to this? (i.e how do the researchers attribute this only to Levo)

We were careful not to make any claims about culturability when discussing our microscopy experiments in the Results section because the dyes can only tell us about loss of cell viability (we discuss viability vs. culturability in the Discussion). The dyes are highly sensitive, meaning that cells that stain SytoxBlue-positive or PI-positive are non-viable, and our data in Fig. S2 demonstrate that the dyes do not affect cell viability under our experimental conditions. In control experiments, we verified that cells that are processed/seeded for time-lapse imaging are viable and able to grow unhindered in our experimental set-up. Here is an example of the viability of *P. aeruginosa* cells on agarose pads made with propidium iodide:

Lines 143-148: How confident are the authors on this statement? This is a conclusion derived not based on direct observations only, but coupled with a deduction, so it may be better included in the discussion

We have rephrased this paragraph to more clearly explain the rationale for the experiment, which is consistent with other plaque assays in the literature (e.g., PMID 25289255):

“To determine whether phages were responsible for the observed lytic phenotypes, we collected cell-free spent media from untreated or Levo-treated cultures to use in plaque assays. We reasoned that if cells in Levo-treated cultures lysed due to phages, then the spent media would contain those lytic phages and lyse untreated *P. aeruginosa* in the plaque assays.”

Lines 150-151: Is this coming from this study? If so it is better to phrase the statement to reflect that

We have added the citations for the *E. coli* phenotypic changes to this sentence, as we had done for the following sentence in the text.

Lines 360-363: Either in the introduction or in the methods, these two media need to be introduced as defined media or something that will justify their use in the experiments

We set up the rationale for using minimal media in the Introduction lines 68-70: “However, many studies on phenotypic responses to antibiotics have been conducted with...stationary-phase cultures that are resuspended in fresh, nutrient-rich media...insufficient to model how pathogens exist in infection sites.” BSM is then introduced in the Results section line 106 as “chemically defined media with succinate as the sole carbon source.” For added clarity, we further explicitly state that they are nutrient-rich or chemically defined media in the methods section.

Lines 366-373: The methods are not clear. Were both genera treated in the same manner except for the two media or not? Why was there a difference? It would be easier if this section is given as a flow chart.

The methods are now shown graphically as diagrams in Fig. S1. The two genera were generally treated in the same manner except for the differences in media used, because *P. aeruginosa* prefers to grow on organic acid carbon sources, whereas *E. coli* prefers glucose, as we note in lines 83-84.

Lines 377-378: If this was done with an E-strip, what were the different dots in Figure 1A? Were replicates done? If replicates were done, state the actual number of replicates done.

The dots in Fig. 1A are showing the individual biological replicates that were conducted for the E-tests (now referred to as “the MIC test strip method”). The figure caption states that the data represent three biological replicates.

Line 384: State how stationary phase was defined? Was it a stagnant OD at two consecutive timepoints or?

Stationary phase was defined by the plateau of OD₆₀₀, as shown in Fig. S7A. For added experimental rigor, we confirmed that *P. aeruginosa* had exhausted succinate, the sole carbon source added to our chemically defined media, at this point. We describe in line 213-214: “The population’s stagnation and succinate consumption indicate that the cultures are in stationary phase.”

Lines 393-396: How many replicates of each tube was taken out at each timepoint?

Only one technical replicate was performed for each experiment (one replicate per tube per timepoint), but each experiment was performed multiple times as biological replicates (testing cultures grown/sampled on different days) for rigor. We explicitly state the number of biological replicates that were performed in our figure captions throughout the manuscript.

Line 427: How many replicates were done for each of the samples?

We show the individual replicates in the figure and state in the caption that at least three biological replicates were done.

Line 430: How was this measured?

The tritiated uridine reagent comes with a Certificate of Analysis that states the radioactivity of the stock solution, so we calculated the volume needed to add 1 μ Ci to each sample.

Line 440: How was the processed?

The “unused” filter was taken from the package and placed directly into scintillation fluid for measuring radioactivity without any processing (no washes, filtration, etc.).

Line 449: This will need to be submitte

Thanks for pointing this out. We have submitted our genomics data to the NCBI Sequence Read Archive under BioProject number PRJNA1034140, set to be released upon manuscript publication or January 1, 2025, whichever comes first. This information is included in the text of the manuscript.

Supplemental Files:

These are not the recommended solvents given in CLSI M100 document. What was the reason for using these solvents? While the methods referred to CLSI guidelines.

The methods specify that we used CLSI guidelines for the broth microdilution method for the thiolutin and antioxidant compounds (thiourea and 2,2'-bipyridine), but the CLSI does not have official guidelines for MIC test strips. We prepared our antibiotic stocks according to the manufacturer’s guidelines, which for Levo and Tobramycin, were the same as the CLSI M100 recommendations. We dissolved Aztreonam and diluted the stocks in DMSO per the manufacturer’s solubility recommendations, which allowed a 40 mg/mL stock to be prepared for our experiments.

It is difficult to understand this in the way it is written. Simply state the concentration range (final) for each of the compounds.

We agree with the reviewer’s suggestion and have changed the concentrations to be listed as a range (rather than just the highest concentration) for clarity.

Re: Spectrum03567-23R1 (Metabolic and Transcriptional Activities Underlie Stationary-Phase *Pseudomonas aeruginosa* Sensitivity to Levofloxacin)

Dear Dr. Wendy W. K. Mok:

Congratulation!

Your manuscript has been accepted, and I am forwarding it to the ASM production staff for publication. Your paper will first be checked to make sure all elements meet the technical requirements. ASM staff will contact you if anything needs to be revised before copyediting and production can begin. Otherwise, you will be notified when your proofs are ready to be viewed.

Sincerely,
Minsu Kim
Editor
Microbiology Spectrum

Reviewer #1 (Comments for the Author):

The Authors have addressed this reviewers concerns adequately

Reviewer #2 (Comments for the Author):

Thank you for painstakingly going to each and every comments and modifying where needed.